# The Importance of the Setting during Sedation for Intrathecal Chemotherapy in Pediatric Oncology Care: A Case Study

**DOI:** 10.3390/healthcare8030314

**Published:** 2020-09-02

**Authors:** Carina Sjöberg, Petra Svedberg, Ing-Marie Carlsson, Jens M. Nygren

**Affiliations:** School of Health and Welfare, Halmstad University, 301 18 Halmstad, Sweden; petra.svedberg@hh.se (P.S.); ing-marie.carlsson@hh.se (I.-M.C.); jens.nygren@hh.se (J.M.N.)

**Keywords:** children, intrathecal chemotherapy, leukemia, pediatric anesthesia, perioperative care

## Abstract

Increasing survival rates for childhood cancer have brought attention to the high level of burden of cancer and its treatment. Improving supportive care for children throughout their cancer trajectory is thus important and could reduce the difficulties related to treatment, including time-consuming treatments and the waiting time associated with treatment procedures. The aim of this study is to describe time intervals and the Propofol dose used during sedation for intrathecal chemotherapy in three different settings. The study is based on retrospective data from repeated treatment sessions recorded in operation planning programs and hospital records in the period 2011–2018 (*n* = 164). Children, 1–12 years old (*n* = 22), undergoing a varying number of treatments, were included in the study. The most crucial finding in this study is that the time from the child’s first meeting with the nurse anesthetist to the induction of sedation is significantly reduced if the procedure is performed in the children’s ward. The study highlights the importance of the setting for sedation for intrathecal chemotherapy when implementing a child-centered approach in pediatric oncology care.

## 1. Introduction

Increasing survival rates of childhood cancer have brought attention to the high level of burden of being treated for cancer as a child [1]. The most common cancer diagnosis among children is leukemia, which is overrepresented in the 1- to 6-year age group, with an incidence peak at 2–4 years of age. Treatment is provided with repeated sessions of intrathecal chemotherapy that lasts, in Sweden, for at least two and a half years, according to Nordic treatment protocols [2]. This treatment forces the children to be hospitalized, interact with unfamiliar people in unknown environments, and undergo unpleasant and painful procedures [3]. A part of the hospitalization process and treatments is having to wait for procedures, which is perceived by children as being one of the worst situations at the hospital [4,5]. Furthermore, being separated from parents during procedures causes anxiety and fear for the youngest children [6].

Being treated for childhood cancer means to be extensively dependent upon others, and the time, place, and activities during the cancer trajectory are, therefore, important for how the children experience their health during illness and treatment. This has, in turn, significant effects on their wellbeing and everyday life [7,8]. Cancer-related pain involves considerable suffering for children during the cancer trajectory and is largely related to procedure-related pain [9]. These challenges can be approached with child-centered care, which takes the children’s preferences into account in order to provide supportive care during procedures and treatment protocols [1,10].

Supportive care, as an integral part of childhood cancer treatment, thus deserves to receive both greater attention and efforts for its improvement [1]. The Multinational Association of Supportive Care in Cancer [11] has defined supportive care in cancer as “the prevention and management of the adverse effects of cancer and its treatment across the cancer continuum”. It includes all care provided to children, in addition to chemotherapy, radiotherapy, surgery, and targeted drugs, and encompasses pharmacological, psychological, and physical treatments [12]. Supportive care should thus be prioritized and further developed in order to improve the management of illness-related distress, patient health outcomes, and experiences during extended hospitalization and long-term treatments [13]. One example of such quality improvement is when children with acute lymphoblastic leukemia are offered supportive care at home after high-dose methotrexate treatment through care in an ambulatory setting [14]. Another example is working in multiprofessional teams to change the location for sedation with Propofol for intrathecal chemotherapy from the operation theatre to the children’s ward. This way of working originated from the team’s perception that the setting and circumstances in the operation theatre were unnecessarily harsh for the children [15]. A qualitative study investigating this quality improvement work showed that the treatment situation became less extensive and more manageable for the children when the location for sedation for intrathecal chemotherapy was moved from the operation theatre to the children’s ward [15]. This latter example of quality improvement work is used as a case for this study. The participants in the multiprofessional team hypothesized that the time for the procedure was shorter when performed in a more peaceful and undisturbed setting at the children’s ward [15]. Our intention with this case study is to quantify and verify if the hypothesized reduction in time used when the procedure is performed at the children’s ward is correct and, if so, if it is significant. The aim of this study is, thus, to describe time intervals and the Propofol dose used during sedation for intrathecal chemotherapy in three different settings.

## 2. Methods

A retrospective case study, with a quantitative design [16], is used to study an existing flexible care procedure. Data from repeated treatment sessions with sedation for intrathecal chemotherapy in three different settings were assessed: (1) the children’s ward, (2) the operating theatre, and (3) the room for minor surgery.

Data were obtained from the operation planning programs and hospital records at a Swedish district hospital for the period 2011–2018. The treatment sessions with sedation were originally only performed in the operating theatre but have, since 2009, been performed in all three different settings. The procedure was initially performed at the children’s ward based on the competence of the health care professionals in service on the current day. Later on, an administrative change in the scheduling for anesthetists resulted in an increase in the number of procedures performed at the children’s ward.

### 2.1. Participants

Inclusion criteria were children aged 1–12 years receiving sedation with Propofol for intrathecal chemotherapy. There were 260 sessions with children receiving intrathecal chemotherapy during the time period, 96 of which were excluded. This was due to the involvement of additional medical measures during the procedure, such as simultaneous bone marrow puncture or additional surgery, or the use of other forms of anesthesia. Another reason for exclusion was if the first author sedated the child. A total of 164 sessions met the inclusion criteria, which entailed that sessions involving 9 girls and 13 boys (*n* = 22) were included in the study (Figure 1). The study was approved by the Regional Ethical Review Board for southern Sweden (registration number: 2016/2). There was no need for informed consent from the included children or their parents since the data did not contain sensitive personal information.

### 2.2. Setting

The entire procedure, namely, the initiation of sedation and intrathecal chemotherapy, was performed in one of the three settings, each of which differed greatly in appearance for the child and had different routines and procedures. The room at the operating theatre had tiled walls, windows, and the standard equipment for operating theatres. The room for minor surgery had tiled walls, windows, and a great deal of medical equipment, an anesthesia machine, but was not as equipped as the operating theatre. The room at the children’s ward was a treatment room adapted for children, decorated with space motifs painted on the ceiling and drapery that hid the medical equipment. Other minor treatments are sometimes performed in conjunction with anesthesia for intrathecal chemotherapy without extending the time or severity of the procedure. Examples of such treatments are replacement of port-a-cath needles, reshaping central iv access, or control of teeth.

### 2.3. The Sedation Procedure and Treatment with Intrathecal Chemotherapy

Sedation for the children in this study was performed by intravenous administration of Propofol in a central line such as a port-a-cath. EMLA (R) was used as a pain reliever [17]. The aim of the sedation is to achieve a level of anesthesia that keeps the child comfortable and allows the performance of the procedure with minimal movement by the child. The sedation was administrated by a specialist anesthetist and a nurse anesthetist in the presence of the child’s parent, who left the room when the child was no longer awake. The children were breathing spontaneously on air throughout the sedation, and pulse oximetry was used for parameters such as heart rate and oxygen saturation. The administration of intrathecal chemotherapy was performed by the pediatrician, with assistance from the pediatric nurse at the children’s ward or the operation nurse in the operating theatre. The procedure was usually limited in time, approximately 15 min, but required the child to lie with a lowered head after the intrathecal injection for at least 30 min in order to optimize the effect of the chemotherapy. When sedation was performed in a treatment room at the children’s ward, the technical equipment was equivalent to that in the operating theatre to ensure a safe procedure.

### 2.4. Data Collection

Data acquired from the operating planning program parameters included the date, place of the procedure, and the time intervals (min.) of the preoperative, intraoperative, and postoperative phases of the procedure (Figure 2). The Propofol dose (mg/kg) used was collected from the anesthesia records.

### 2.5. Statistical Analysis

The analyses were carried out based on the assumption that the setting during sedation for intrathecal chemotherapy is important for the outcome in terms of time intervals and Propofol dose used. Statistical comparison of data on the procedure from the different settings was made using nonparametric tests since there was no normal distribution of data; the Kruskal–Wallis test and posthoc analyses, including the Mann–Whitney test with Bonferroni correction, were applied. Statistical comparisons of the children’s age distribution in the different settings were made using the Kruskal–Wallis test. Descriptive statistics were used to characterize the population using frequencies and mean. The statistical analyses were performed using IBM Corp.Released 2016. IBM SPSS Statistics for Windows, Version 26.0. (IBM Corp., Armonk, NY, USA).

## 3. Results

Data from 164 procedures involving 22 children were collected (Table 1). The mean number of procedures per participant was 7.4, ranging from 1 to 17. The procedures were unevenly distributed between the children’s ward (62.8%), the operating theatre (23.8%), and the room for minor surgery (13.4%). There was no statistically significant difference in the age distribution of the children in the three different settings (*p* = 0.269).

An obvious outcome from providing the procedure at three different settings was preoperative time reduction since transportation to the operating theatre could be avoided. The preoperative starting time interval was thus significantly shorter for the children undergoing the procedure at the children’s ward (*p* = 0.001). The next time interval, the preoperative waiting time, did not significantly differ between the three settings, indicating that when the process for the procedure had been initiated, the waiting time before the actual procedure did not differ between the settings. When the preoperative starting time interval was added with the preoperative waiting time (named preoperative time), this total preoperative time was significantly shorter for the children undergoing the procedure at the children’s ward (*p* = 0.001).

The intraoperative waiting time interval was also significantly shorter for the children undergoing the procedure at the children’s ward (*p* = 0.001), as was the combination of the pre- and intraoperative waiting times (*p* = 0.001). The time intervals, from the start of sedation until the administration of intrathecal chemotherapy was completed, as well as the actual time for the administration of intrathecal chemotherapy, were not significantly different between the settings (Table 2). No significant differences were found in the Propofol dose (mg/kg) used between the settings (Table 3), which indicates that confounders affecting the Propofol dose had no significant role in this case study.

## 4. Discussion

This study uses a particular procedure during the pediatric cancer treatment trajectory as a case with the objective of investigating time usage and Propofol dose used in different settings for children’s cancer treatment. The results present both the expected findings of overall shorter time usage if the procedure is performed in the children’s ward and the significant reduction of waiting time for children during the procedure. The results highlight that shorter time usage during intrathecal chemotherapy treatment can help to generate a child-centered approach that can contribute to more supportive care. The study objectively confirms experience-based tacit knowledge within a multiprofessional team performing this treatment, studied in a previous project [15].

The short distance to the treatment room in the children’s ward means that a very limited amount of time is needed to transport the child prior to the start of the intrathecal chemotherapy procedure and, thus, that the procedure also impacts the child’s life situation to a lesser extent. However, this study also provides objective confirmation of a significant reduction in the time from when the child first meets the nurse anesthetist until the induction of sedation if the procedure is performed in the children’s ward.

This shorter time usage can be of major importance for how the children experience the procedure as such, but also for their whole care situation [15,18]. The hospital environment, having to wait for procedures, and being separated from their parents are the major causes of anxiety and fear among children treated for cancer [4,8,19]. These circumstances are affected by where the procedure for chemotherapy is to be performed. In addition to being an inconvenience for the child, transportation to the operating theatre also entails that the child is exposed to other care environments, which can generate a major risk for these seriously ill children [15,20]. Optimizing the selection of setting for the procedure thus becomes a tangible way of working towards reducing the anxiety and fears of the children and making it possible to reduce the negative effects from the procedure as a whole. These include postoperative pain and late effects such as sleep difficulties and nightmares [18,21]. This study shows, indirectly, that by performing intrathecal chemotherapy at the children’s ward, the children can remain with their family for a longer period of time prior to the procedure and return to them more quickly, thus reducing the intrusion of the procedure in their everyday lives [15]. Disruptions in family cohesion impact the quality of life for families during their children’s cancer trajectory [14] and sustaining family rituals functions as a resource to meet the psychosocial needs of children with cancer and their parents during the children’s cancer trajectory [22]. This highlights the importance of reflecting on how to adapt protocols and procedures to maintain bonds between family members during cancer treatment in the hospital. Furthermore, by reducing the time for the procedure by performing it in the children’s ward, conditions for the children to play are improved both in terms of time and access to toys and playrooms. The importance of play for children when they are in the hospital cannot be overestimated as it provides opportunities for distraction, rest, and coping with difficulties [23,24,25].

The intention of this case study is not generalization. However, this type of study can have high external validity and can contribute to knowledge that can be used in other contexts [26]. The limitations of the study mainly relate to it containing limited data, thus making it vulnerable to confounders affecting the result. The main confounder, in this case, is the lack of standardized routines for how to make decisions on which setting to use for this procedure. However, no uneven distribution of where the treatment took place between the different settings was seen based on age or the timing of the procedure in relation to the children’s cancer treatment trajectory. However, a standardized routine in this setting would probably have limitations because each procedure requires the individual medical assessment of a seriously ill child to determine the optimal setting for the procedure on each day. Due to the limited number of individuals in the analysis, the findings must be interpreted with caution due to the outliers in the dataset in the form of extremes in the total time required for the procedure. These were presumably due to difficulties in the documentation of the operating planning programs and simultaneously performance of the procedure, and these procedures thus need to be both improved and studied. The results from this case study can thus not be considered absolute and generalizable but need to be focused on in future research in order to achieve reliability [27]. It would be of particular interest to investigate the children’s perspectives on aspects of the usage of their time during pediatric cancer treatment.

In conclusion, sedation for intrathecal chemotherapy in the children’s ward provides care, which corresponds to a greater degree to the needs of children by eliminating time to the start of the procedure, reducing the waiting time during the procedure, and thus the total procedure time. However, there is a need for further studies, with children undergoing the procedure in other settings at other hospitals in other regions, to further understand how the setting during sedation for intrathecal chemotherapy in pediatric oncology care can contribute to a more child-centered approach for the procedure.

### Implications for Clinical Practice

Children undergoing cancer treatment face many multifaceted stressors, and even small adjustments of the treatment can be important to reduce stress for the children. By changing the location for sedation for intrathecal chemotherapy from the operation theatre to the children’s ward, the healthcare professionals enabled a reduction of time, prior and during the procedure. The selection of setting for the procedure can thus be used as part of supportive care towards reducing the anxiety and fears of the children.

## 5. Conclusions

The study uses the implementation of a specific procedure at a regional hospital as a case to illustrate the importance of the setting for providing a child-centered approach during sedation for intrathecal chemotherapy. The setting for procedures like intrathecal chemotherapy affects the use of children’s time during a cancer trajectory and is, therefore, a relevant area for improvement to enhance supportive care. The procedure is carried out in a similar manner and with the same prerequisites at many hospitals. The result of this study is thus interesting, but more research is required in other contexts to be able to inform the general improvement of supportive care for this patient group. The result can presumably be transferred to other children undergoing strenuous treatments and examinations in other high technological settings.

## Figures and Tables

**Figure 1 healthcare-08-00314-f001:**
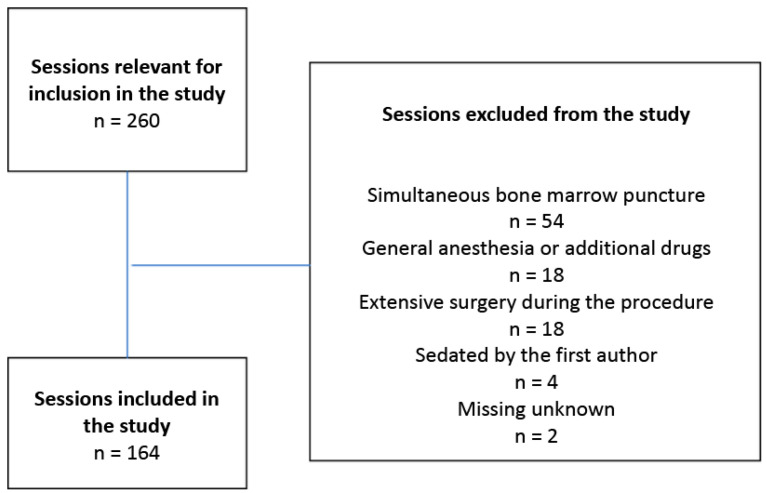
Selection process of sessions and reasons for exclusion.

**Figure 2 healthcare-08-00314-f002:**
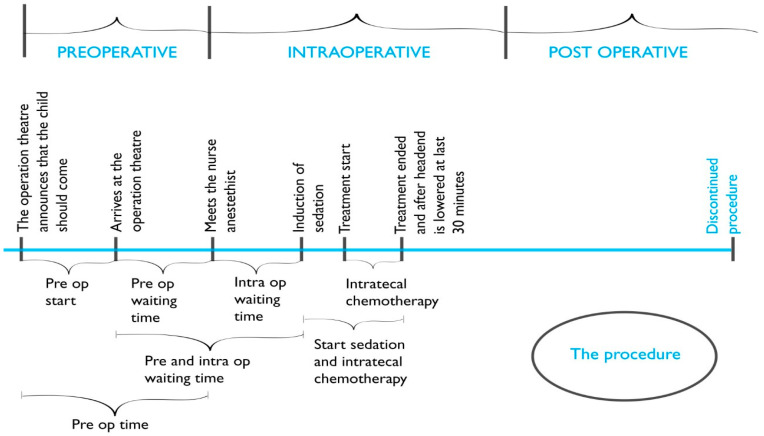
The three phases of the procedure for intrathecal chemotherapy and time intervals investigated.

**Table 1 healthcare-08-00314-t001:** Characteristics of the 22 patients (164 procedures).

Age-Ranges	Procedures No.	(%)
0–3	55	33.5
4–7	89	54.3
8–11	15	9.2
≤12	5	3.0
Total	164	100

**Table 2 healthcare-08-00314-t002:** Time intervals (min), median (range).

Setting	Operation Theatre	Children’s Ward	Room for Minor Surgery	*p*-Value *
No of procedures	39	103	22	
Preop start	5 (0–40)	0 (0–30)	8 (0–75)	0.001 **
Preop waiting time	2 (1–20)	1 (1–60)	1 (1–17)	0.114
Pre op time	12 (0–43)	5 (0–60)	10 (0–75)	0.001 ***
Intraop waiting time	10 (1–29)	5 (1–30)	10 (1–40)	0.001 ****
Pre- and intraop waiting time	13 (2–40)	8 (2–70)	15.5 (2–50)	0.001 *****
Start of sedation and intrathecal chemotherapy	16 (6–45)	15 (2–45)	17.5 (5–53)	0.304
Intrathecal chemotherapy	15 (2–85)	12 (1–45)	15 (5–50)	0.151

* Kruskal–Wallis test and posthoc Bonferroni of the three different settings: operation theatre (1), children’s ward (2) and room for minor surgery (3). ** 1 vs. 2 = 0.001 and 2 vs. 3 = 0.012; *** 1 vs. 2 = 0.001 and 2 vs. 3 = 0.009; **** 1 vs. 2 = 0.003 and 2 vs. 3 = 0.04; ***** 1 vs. 2 = 0.003 and 2 vs. 3 = 0.02.

**Table 3 healthcare-08-00314-t003:** Propofol dose (mg/kg), median (range).

Setting	Operation Theatre	Children’s Ward	Room for Minor Surgery	*p*-Value *
No. of Procedures				
Valid	33	91	19	
Missing	6	12	3	
Dose	7.7 (3.4–14.2)	8.4 (2.9–20.4)	7.6 (3.2–15.9)	0.205

* Kruskal–Wallis test.

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
