# Peer review of "The Importance of the Setting during Sedation for Intrathecal Chemotherapy in Pediatric Oncology Care: A Case Study"

_healthcare, 2020, doi:10.3390/healthcare8030314_

Round 1

Reviewer 1 Report

This manuscript entitled “The importance of the setting during sedation for intrathecal chemotherapy in pediatric oncology care: a case study” by Carina Sjöberg et al. would like to compare the operation time for the pediatric patients undergoing intrathecal chemotherapy in three different places, Operation theatre, Children’s ward, Room for minor surgery. They found waiting time could be reducing if procedure was performed in ward, therefore, they suggest the importance of the setting for sedation for intrathecal chemotherapy for implementing a child centered approach in pediatric oncology care.

These results could be expected from experience of general practice so nothing novel was found from this study. This study is to prove the concept. The authors merely evaluate the times and did not perform objective evaluation from patients to prove this could improve the quality of the children. Authors emphasize this can enhance supportive care but no associated results were found. In addition, this study includes 164 procedures involving 22 children. The authors did not evaluate the operative time in different place within the same patient.

Some minor comments:

  1. Line 4 of introduction: intratecal à intrathecal

  1. The format of decimal point may be wrong.

  1. This study enrolled patients aged 1-12 years. How can the patients be older than 12 (table 1)?

  1. Missing data in table 2?

  1. Sessions were additional drugs were used or the first author sedated the child were also excluded. à grammar errors and unclear sentence.

Reviewer 2 Report

Tha paper Sjöberg, C. is interesting but the representation of the paper should really be improved to make it interesting to the general readers. 164 procedures were carried over among the patients but it is not absolutely clear to me what are these procedures?

Also, not clear that the amount of sedation given in the lower part of the Table 2 is pretty similar. Then, what is the point the author is trying to make?

The notion is quite innovative and can be interesting to different other fields as well. But, the authors should improve the introduction and discussion to convince the general readers. 

Reviewer 3 Report

As a general comment: it is clear that the manuscript was not written by a native English speaker. Some sentences miss the subjects, others are not well constructed. A serious revision is required.

Also, it appears that the manuscript lack a solid outline.

In addition, the main limitation is related to the fact that there was not an a priori decision of destination to treat (theatre, ward, etc). The authors should highlight this issue in the discussion (more in depth as they have already done).

Abstract: Remove the part of statistical analysis. Is not important in the abstract.

Methods: Please explain better the inclusion/exclusion criteria. Maybe create an ad hoc paragraph.

Results:

1) "These were unevenly distributed between the children ́s ward (62,8%), the operating theatre (23,8%), and the room for minor surgery (13,4%)." What is the subject of these? Because these sentence is unclear if referred to the previous one.

2) Please report correctly p-values: p=0.001 and not p=.000.

Round 2

Reviewer 1 Report

Thanks for author's great work on improvement of this manuscript.

Some grammatical errors are still noted in the manuscript so English editing is necessary before publication. Otherwise, no more comments I can provide.

For example:

“The most crucial finding in this study is that the time from the child’s first meeting with the nurse anesthetist to the induction of sedation is started is significantly reduced if the procedure is performed on the children’s ward. " in Abstract. This sentence contains lots of "is" and "is started" may be deleted.

Reviewer 2 Report

The authors have significant changes to the manuscript and it can be accepted in the current format.